# GraphVF: Controllable Protein-Specific 3D Molecule Generation with Variational Flow

## Abstract

Designing molecules that bind to specific target proteins is a fundamental task in drug discovery. Recent generative models leveraging geometrical constraints imposed by proteins and molecules have shown great potential in generating protein-specific 3D molecules. Nevertheless, these methods fail to generate 3D molecules with 2D skeletal curtailments, which encode pharmacophoric patterns essential to drug potency and synthesizability. To cope with this challenge, we propose GraphVF, which integrates geometrical and skeletal restraints into a variational flow framework, where the former is captured through a normalizing flow transformation and the latter is encoded by an amortized factorized Gaussian. We empirically verify that our method achieves state-of-the-art performance on protein-specific 3D molecule generation in terms of binding affinity and some other drug properties. In particular, it represents the first *controllable* geometry-aware, protein-specific molecule generation method, which enables creating binding 3D molecules with specified chemical sub-structures or drug properties.

## 1 Introduction

The *de novo* design of synthetically feasible drug-like molecules that bind to specific protein pockets is a crucial yet very challenging task in drug discovery. To cope with such challenge, there has been a recent surge of interest in leveraging deep generative models to effectively searching the chemical space for molecules with desired properties. These machine learning models typically encode the chemical structures of molecules into a low-dimensional space, which then can be optimized and sampled to generate potential 2D or 3D molecule candidates (Jin et al., 2018; Shi et al., 2020; Zhu et al., 2022; Hoogeboom et al., 2022). Along this research line, a more promising direction has also been explored recently: generating 3D molecules that bind to given proteins.

Such binding 3D molecule generation is fundamentally important because binding in fact mainly facilitates the functionalities of drugs. Fortunately, leveraging autoregressive models to generate drug molecules (i.e., ligands) directly based on the 3D geometry of the binding pocket has shown promising potential (Luo et al., 2021; Peng et al., 2022; Liu et al., 2022). These methods explicitly capture the fine-grained atomic interactions in the 3D space, and produce ligand poses that can directly fit into the given binding pocket. Nevertheless, two critical issues remains unsolved for these existing geometric approaches: 1) effective encoding and sufficient preservation of pharmacophoric structural patterns in the ligand candidates, and 2) controllable ligand generation that aims at specified drug properties or sub-structures. The former prevents generating ligands that seem geometrically plausible, yet structurally invalid or pharmacophorically impotent; the later dominates the synthesibility and the practical usefulness of the drugs. We further elaborate them next.

In practice, it is extremely valuable to keep track of the pharmacophoric patterns in the existing ligands, which indeed determines a ligand's bio-chemical activities and binding affinity to a large extent (Wermuth et al., 1998). Consider, for example, the molecules of serotonin (a benign neurostransmiter) and N,N-Dimethyltryptamine (DMT, a famous hallucinogen). As can be seen in Figure 5a from Appendix E, serotonin and DMT share a large common bulk of their structures, which both possess an indole and an ethylamine group, but differ enormously in their neural activities. In fact, the extra Methyl groups in DMT's $NHMe_2$ are pharmacophoric, inducing an attractive charge interaction with Asp-231 (Gomez-Jeria & Robles-Navarro, 2015). This pharmacophoric feature gives rise to DMT's binding affinity with the $5\text{-}HT_{2A}$ binding site and produces hallucination.

Table 1: Comparison among representative molecular generative methods.

| Name | Generative Model | Moleculer Encoding | | | Controlled Generation | | |
|------|------------------|------|------|----------|--------------|---------------|--------------|
| | | Atom | Bond | 3D Coord. | Ligand Struc. | Receptor Conf. | Generic Prop. |
| EDM | Diffusion | ✓ | - | ✓ | - | - | ✓ |
| DMCG | VAE | ✓ | ✓ | ✓ | ✓ | - | - |
| JT-VAE | VAE | ✓ | ✓ | - | ✓ | - | - |
| GraphAF | Autoregressive Flow | ✓ | ✓ | - | - | - | - |
| GraphBP | Autoregressive Flow | ✓ | - | ✓ | - | ✓ | - |
| Pocket2Mol | Spatial Autoregression | ✓ | ✓ | ✓ | - | ✓ | - |
| GraphVF | Variational Flow | ✓ | ✓ | ✓ | ✓ | ✓ | ✓ |

Such observations suggest that effectively enforcing pharmacophoric patterns in ligands is critical for binding.

Equally important, controlling molecular properties like solubility, polarizability and heat capacity are instrumental to drug quality. This is to make sure that the synthesized drug molecules have good exposure, *e.g.* absorption/distribution/metabolism/excretion (ADME) *in vivo*, and thus, sufficient efficacy in clinical trials (Egan, 2010). It is worth noting that, although recent diffusion models like EDM (Hoogeboom et al., 2022) have been popular for their capability to perform controlled generation on these properties, performing such control while being pertinent to a given pocket structure for binding remains under-explored by previous works.

To address the aforementioned two issues, we propose GraphVF, a protein-aware molecule generation framework that integrates both geometrical and skeletal constraints, aiming at controlling over the structure and property of the generated ligands. To attain this goal, we leverage flow-based architecture that combines amortized variational inference (Zhang et al., 2018) and autoregressive normalizing-flow generation. In specific, global structure of the drug ligand is organized as a junction tree (Jin et al., 2018), and fine-grained geometrical context of the protein receptor is encoded via a valence-aware E(3)-GNN. These two constraints are integrated into a variational flow architecture, where the former enforces the variational distribution globally, while the latter administers the flow transformations autoregressively.

We show empirically that, GraphVF generates drug molecules with high binding affinity to the receptor proteins, with or without the aid of reference ligands, outperforming state-of-the-art methods in terms of binding affinity and some other drug properties. More importantly, GraphVF exposes a clean-cut interface for imposing customized constraints, which is extremely useful in practice for controlling the sub-structure and bio-chemical property of generated drug ligands. To specify what our proposed model can actually do, we compare GraphVF with several representative models for molecule generation in Table 1.

Our main contributions are summarized as follows.

- We devise a novel variational flow-based framework to seamlessly integrate geometrical and skeletal restraints to improve protein-specific 3D molecule generation.

- We show the first method that enables generating 3D molecules with specified chemical sub-structures or bio-chemical properties.

- We empirically demonstrate our method's superior performance to state-of-the-art approaches on generating binding 3D molecules.

## 2 RELATED WORK

**Non-Protein Specific Molecule Generation** Different generative techniques have been applied to the task of molecular generation, including Variational Autoencoders (VAEs) (Kingma & Welling, 2013), Diffusion Models (Sohl-Dickstein et al., 2015), Normalizing Flows (NFs) (Dinh et al., 2016), and Autoregressive Models (Van Oord et al., 2016). The line of work is usually context-free, aiming to produce high-quality molecules from scratch, or to render reasonable 3D conformations of given molecules. For example, JT-VAE (Jin et al., 2018) generates molecular graphs with the guidance of a tree-structured scaffold over chemical substructures. GraphAF (Shi et al., 2020) uses a flow-based model to generate atoms and bonds in an autoregressive manner. DMCG (Zhu et al., 2022) and EDM (Hoogeboom et al., 2022) leverage equivariant diffusion or iterative sampling and de-noising to generate 3D conformations from 2D structures. Unlike these methods, our approach aims at generating molecules that bind to given 3D protein pockets.

**3D Molecule Generation for Target Protein Binding**   With the wide availability of large-scale datasets (Francoeur et al., 2020; Li et al., 2021) for target protein binding, recent works have been able to generate drug ligands directly based on the 3D geometry of the binding pockets. For example, Pocket2Mol (Peng et al., 2022) leverages a spatial-autoregressive model; it directly models the *p.d.f.* for atom occurrence in the 3D space as a Gaussian mixture (GMM), and then iteratively places the atoms from the learned distribution until there is no room for new atoms. GraphBP (Liu et al., 2022), an autoregressive model, retains good model capacity via normalizing flow; variables are randomly sampled from a compact latent space, before they are projected into the chemical space by an arbitrarily complex flow transformation. Despise their promising potential, these methods ignore the topological organization of the drug ligand itself, as well as the structural patterns and pharmacophoric features embodied in it. As a result, existing methods tend to generate ligands that seem geometrically plausible, yet structurally invalid or pharmacophorically impotent. Our approach here aims to address this problem. Also, our method enables controllable molecule generation, facilitating generating drug ligand candidates with specified chemical sub-structures or drug properties.

## 3   PRELIMINARIES

Our proposed method leverages an autoregressive flow strategy to generate binding molecules.

### 3.1   AUTOREGRESSIVE FLOW MODELS

Given a prior distribution $p_Z$, a flow model (Dinh et al., 2014; Rezende & Mohamed, 2015; Weng, 2018) is defined as an invertible parameterized function $f_\theta : \mathbf{z} \in \mathbb{R}^D \to \mathbf{x} \in \mathbb{R}^D$, where $\theta$ represents the parameters of $f$, and $D$ is the dimension for $\mathbf{z}$ and $\mathbf{x}$. This maps the latent variable $\mathbf{z} \sim p_Z$ to the data variable $\mathbf{x}$, and the log-likelihood of $\mathbf{x}$ is calculated as

$$\log p_X(\mathbf{x}) = \log p_Z\left(f_\theta^{-1}(\mathbf{x})\right) + \log\left|\det\frac{\partial f_\theta^{-1}(\mathbf{x})}{\partial \mathbf{x}}\right|. \tag{1}$$

Autoregressive flow model (Papamakarios et al., 2017) formulates a flow model with an autoregressive computation to enable easy Jacobian determinant computation. Let $\mathbf{x}_i$ be the $i$-th component of $\mathbf{x}$ and $\mathbf{x}_i$ conditions on $\mathbf{x}_{1...i-1}$. The inverse function $f_\theta^{-1}$ is then defined as follows:

$$\mathbf{x}_i = \sigma_i(\mathbf{x}_{1...i-1}) \odot \mathbf{z}_i + \mu_i(\mathbf{x}_{1...i-1}), \quad i = 1...D, \tag{2}$$

where $\odot$ denotes element-wise multiplication, $\sigma_i(\cdot) \in \mathbb{R}$ and $\mu_i(\cdot) \in \mathbb{R}$ are non-linear functions of $\mathbf{x}_{1...i-1}$. Doing so, we can effectively calculate the follows:

$$\mathbf{z}_i = \frac{\mathbf{x}_i - \mu_i}{\sigma_i}, \quad \det\frac{\partial f_\theta^{-1}(\mathbf{x})}{\partial \mathbf{x}} = \prod_{i=1}^{D}\frac{1}{\sigma_i}. \tag{3}$$

### 3.2   PROBLEM FORMULATION AND NOTATIONS

Given a specific protein receptor, our task is to generate a ligand molecule that binds effectively to it. Here, proteins and ligands are represented as graphs in the 3D geometric space. Node features include atom type $a$ and position $r$, while edge feature involves bond type $b$. For training, we are given pairs of protein $\mathcal{P}$ and ligand $\mathcal{R}$ in their binding poses. For generation, we are given protein targets $\mathcal{P}$ to generate drug ligands that bind tightly to them. Consider a given protein-ligand pair with $M$ and $N$ atoms respectively. We denote the protein as $\mathcal{P} = (\tilde{V}, \tilde{E})$, where $\tilde{V} = \{(\tilde{a}_i, \tilde{r}_i)\}_{i=1}^{M}$ and $\tilde{E} = \{\tilde{b}_{ij}\}_{i,j=1}^{M}$, and the ligand as $\mathcal{R} = (V, E)$, where $V = \{(a_i, r_i)\}_{i=1}^{N}$ and $E = \{b_{ij}\}_{i,j=1}^{N}$. With a slight abuse of notation, the 2D topology of the ligand structure is also referred to as $\mathcal{R}$.

## 4   THE PROPOSED METHOD

In this section, we first introduce how we encode both the ligand scaffold (Section 4.1) and the protein-ligand geometry information 4.2). This is followed by Section 4.3, which presents how these two pieces of knowledge are used to generate a binding molecule autoregressively. Finally, in Section 4.4, we detail how we cope with three challenges arising in Section 4.3 through a variational flow model.

## 4.1 LIGAND SCAFFOLDS ENCODING

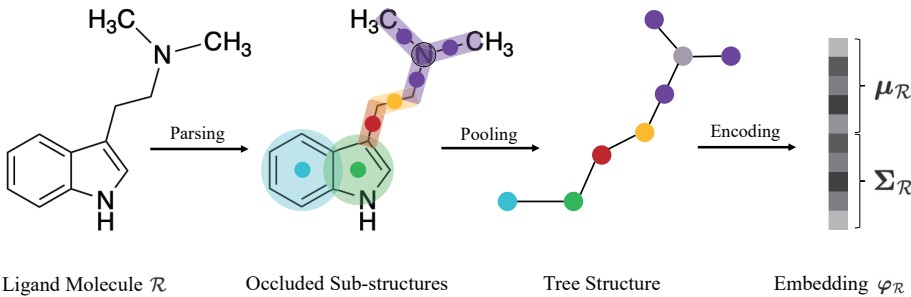

Figure 1: Ligand molecule $\mathcal{R}$ (e.g. DMT) is first parsed into a compilation of canonical substructures, then pooled into a junction tree structure, and finally encoded into $\varphi_{\mathcal{R}} = (\boldsymbol{\mu}_{\mathcal{R}}, \boldsymbol{\Sigma}_{\mathcal{R}})$.

Inspired by (Jin et al., 2018), we extract the coarse-grained structural patterns of the ligand scaffold in a fragment-driven approach. The whole procedure is illustrated in Figure 1, and the detail is discussed next.

First, the ligand molecule $\mathcal{R}$ is parsed into a compilation of occluded canonical sub-structures, according to a set of pre-defined vocabulary (more details are in Appendix A.1). Next, the resulting graph structure is pooled into a junction tree structure. Structural patterns are well exposed in the junction tree, where each node represents a sub-structure. Finally, the structral encoding $\varphi_{\mathcal{R}} = (\boldsymbol{\mu}_{\mathcal{R}}, \boldsymbol{\Sigma}_{\mathcal{R}})$ is derived from the tree structure via a Gated Recurrent Unit (GRU) (Chung et al., 2014) adapted for tree message passing. In particular, $\boldsymbol{\mu}_{\mathcal{R}}$ and $\boldsymbol{\Sigma}_{\mathcal{R}}$ are equally-sized dense vectors, which parameterize the amortized variational distribution to be discussed in Sections 4.3 and 4.4. Note that $\varphi_{\mathcal{R}}$ is merely the shorthand symbol for the concatenation of $\boldsymbol{\mu}_{\mathcal{R}}$ and $\boldsymbol{\Sigma}_{\mathcal{R}}$. It should also be emphasized that GRU encoder and $\varphi_{\mathcal{R}}$ are not fixed: they are trained in an **end-2-end** manner alongside the whole model to avoid deviations from the training objective. Refer to Appendix A.2 for implementation details about the GRU architecture.

## 4.2 GEOMETRY GRAPH ENCODING

Equivariant graph neural networks like SchNet (Schutt et al., 2017) and EGNN (Satorras et al., 2021) have become a routine component in this receptor-based line of work, which are essential for encoding molecular features with roto-translational equivariance. Atoms around the binding pocket are organized into a $k$NN/radius graph, based on their euclidean distance in the 3D space. This distance-based approach is appropriate for modeling non-covalent interactions like hydrogen bonds and hydrophobic interactions, but remains inadequate for modeling covalent bonds. Bond lenghs are known to be characteristic, e.g. C≡N 1.16 Å, C=C 1.34 Å (Lide, 2012). Explicitly incorporating bond types during message passing is thus beneficial for better perception of atomic interactions and more reasonable delineation of molecular structures.

To achieve this goal, we devise **Echnet**, an SE(3)-equivariant graph neural network specially tailored for bond type message passing in the 3D geometric setting, which is formulated as follows:

$$\mathbf{h}_i^{(0)} = \text{Emb}(a_i) \tag{4}$$

$$\mathbf{m}_{ij} = \text{concat}\left\{\text{Erbf}(||r_i - r_j||), \text{Emb}(b_{ij})\right\} \tag{5}$$

$$\mathbf{h}_i^{(l)} = \mathbf{h}_i^{(l-1)} + \sum_{k \in N(i) \setminus j} \mathbf{h}_k^{(l-1)} \odot \Phi^{(l)}(\mathbf{m}_{ki}), \quad l = 1, ..., L \tag{6}$$

where $\Phi$ is the feed-forward neural network, $\text{Erbf}(\cdot)$ is radial basis function (Liu et al., 2022), $\text{Emb}(\cdot)$ is the embedding layer, $\text{concat}(\cdot)$ is the concatenation of two vectors, and $\Phi$ is the feed-forward neural network. $L$ is the number of convolution layers, $\mathbf{h}_i^{(l)}$ stands for the encoding of atom $i$ at the $l^{\text{th}}$ convolution layer, and $\mathbf{m}_{ij}$ is message passed from atom $i$ to $j$.

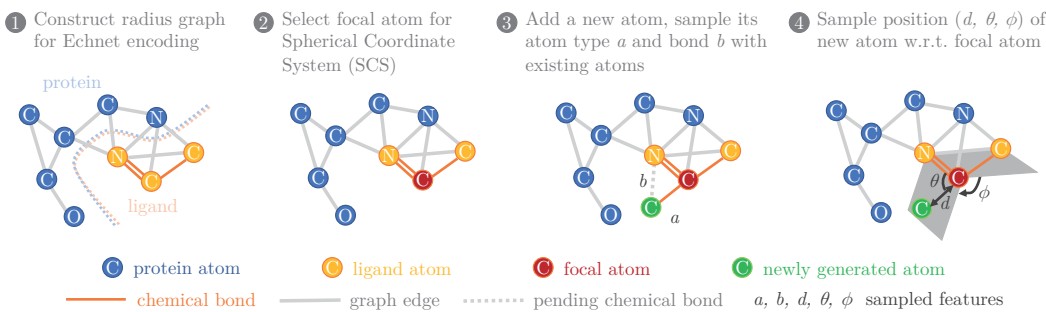

Figure 2: Generation procedure of GraphVF. Atoms are added autoregressively, whose types, bonds and positions are sampled from prior distribution $\mathcal{N}(\boldsymbol{\mu}_\rho, \boldsymbol{\Sigma}_\rho)$ and predicted via normalizing flow.

## 4.3 BINDING LIGAND GENERATION

We formalize the procedure of new ligand generation as a Markovian sampling process, where atoms and bonds are autoregressively added according to the intermediary state at the binding site. The generation process at step $i = 4$ is illustrated in Figure 2. In particular, we provide a prior distribution $\mathcal{N}(\boldsymbol{\mu}_\rho, \boldsymbol{\Sigma}_\rho)$ to control certain desired traits of the generated ligand. The implication and derivation of $\mathcal{N}(\boldsymbol{\mu}_\rho, \boldsymbol{\Sigma}_\rho)$ will be discussed in Section 4.4. When not specified, the prior is set to $\mathcal{N}(\mathbf{0}, \boldsymbol{I})$ by default. We now elucidate the generation process at step $i$.

Firstly, we construct radius graph $\mathcal{G}_i$ based on protein graph $\mathcal{P}$ and ligand sub-graph $\mathcal{R}_{1:i-1}$:

$$\mathcal{G}_i = \tau(\mathcal{P} \cup \mathcal{R}_{1:i-1}) \tag{7}$$

where the radius operator $\tau(\cdot)$ adds edges (of bond order 0) to neighboring atoms within radius $\tau$. In particular, at generation step 1, when no ligand atoms have yet been generated, $\mathcal{G}_i$ is simply $\tau(\mathcal{P})$. Echnet outputs the encoding of each atom in both protein and ligand:

$$\tilde{\mathbf{e}}_{1:M}, \mathbf{e}_{1:N-1} = \text{Echnet}(\mathcal{G}_i) \tag{8}$$

Secondly, we select a focal atom $f_i$ with the focal classifier. Except in the first step, $f_i$ is only selected from the drug ligand. Based on $f_i$ and two of its nearest neighbors, we construct a spherical coordinate system (SCS), transforming Cartesian coordinates into polar coordinates $(d, \theta, \phi)$.

Finally, we add a new atom to the drug ligand via sequential generation of its atom type $a_i$, bindings with existing atoms $b_{1:i-1,i}$ and position $r_i = (d_i, \theta_i, \phi_i)$, in order to better capture the underlying dependencies (Liu et al., 2022). The prior random variables are sampled from $\mathcal{N}(\boldsymbol{\mu}_\rho, \boldsymbol{\Sigma}_\rho)$, including $\mathbf{z}_i^{(\text{node})}$, $\mathbf{z}_{1:i-1,i}^{(\text{bond})}$ and $\mathbf{z}_i^{(\text{pos})}$. The priors are then consecutively projected to the 3D geometric space via flow transformation $\mathcal{F}_i$:

$$\mathbf{x}_i^{(\text{node})}, \mathbf{x}_{1:i-1,i}^{(\text{bond})}, \mathbf{x}_i^{(\text{pos})} = \mathcal{F}_i\left(\mathbf{z}_i^{(\text{node})}, \mathbf{z}_{1:i-1,i}^{(\text{bond})}, \mathbf{z}_i^{(\text{pos})}; \mathbf{e}_{1:N-1}\right) \tag{9}$$

Specifically, the flow transformation $\mathcal{F}_i$ is parameterized as follows:

$$\mu_i^{(\text{node})}, \sigma_i^{(\text{node})} = \text{Node-MLP}(\mathbf{e}_{f_i}) \tag{10}$$

$$\mathbf{x}_i^{(\text{node})} = \sigma_i^{(\text{node})} \odot \mathbf{z}_i^{(\text{node})} + \mu_i^{(\text{node})} \tag{11}$$

$$\mu_{1:i-1,i}^{(\text{bond})}, \sigma_{1:i-1,i}^{(\text{bond})} = \text{Bond-MLP}(\mathbf{e}_{1:i-1}, \mathbf{x}_i^{(\text{node})}) \tag{12}$$

$$\mathbf{x}_{1:i-1,i}^{(\text{bond})} = \sigma_{1:i-1,i}^{(\text{bond})} \odot \mathbf{z}_{1:i-1,i}^{(\text{bond})} + \mu_{1:i-1,i}^{(\text{bond})} \tag{13}$$

$$\mu_i^{(\text{pos})}, \sigma_i^{(\text{pos})} = \text{Position-MLP}(\mathbf{e}_{f_i}, \mathbf{x}_i^{(\text{node})}, \mathbf{x}_{1:i-1,i}^{(\text{bond})}) \tag{14}$$

$$\mathbf{x}_i^{(\text{pos})} = \sigma_i^{(\text{pos})} \odot \mathbf{z}_i^{(\text{pos})} + \mu_i^{(\text{pos})} \tag{15}$$

where $\odot$ denotes element-wise multiplication, and $\mathbf{x}_i^{(\text{node})}$, $\mathbf{x}_{1:i-1,i}^{(\text{bond})}$, $\mathbf{x}_i^{(\text{pos})}$ are the vectorized representation of atom type, bond type and position, and $\sigma, \mu$ are parameters for flow transformation. The sequential dependency between $a, b, d$ is embodied in Equations 12 and 14, where new atom/bond

types that have just been generated are immediately used to parameterize $\sigma$ and $\mu$ of the next flow transformation.

Thus, we have rendered all the sampled features $a_i, b_{1:i-1,i}, d_i, \theta_i, \phi_i$ from step $i$, and successfully generate the new atom and its associated bonds. We go on with this iteration, until the focal classifier reports that no atom is eligible for $f_i$, and the generation procedure is called to an end. Algorithm 2 from Appendix B explains the generation algorithm in more detail.

## 4.4 VARIATIONAL FLOW TRAINING

Though self-contained, the generation procedure in Section 4.3 puts forward three critical technical challenges nonetheless:

1) Integration of both binding pocket geometry and ligand structural patterns;
2) Generation of molecules with high binding affinity, even without reference ligands;
3) Controllable generation interface for customized bio-chemical constraints.

We propose **variational flow**, a dedicated training framework, to solve the above challenges. As is illustrated in Figure 3, the term is coined according to the two key components in the framework: the amortized variational distribution $\mathcal{N}(\boldsymbol{\mu}_\mathcal{R}, \boldsymbol{\Sigma}_\mathcal{R})$, and the invertible flow function $\mathcal{F}_i$. We proceed to discuss the different components of this training framework.

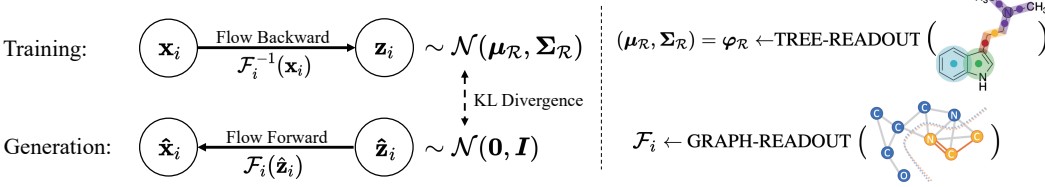

Figure 3: Comparison of a single training/generation step in the variational flow framework

**Encode 2D structure:** The primary difference between training and generation is that, the complete structure of the reference ligand is only available during training. Therefore, for each drug-ligand pair, we use the junction tree encoder (Section 4.1) to encode the reference ligand's structral pattern as $\varphi_\mathcal{R} = (\boldsymbol{\mu}_\mathcal{R}, \boldsymbol{\Sigma}_\mathcal{R})$ **on the fly**, which parameterizes the variational distribution $\mathcal{N}(\boldsymbol{\mu}_\mathcal{R}, \boldsymbol{\Sigma}_\mathcal{R})$.

**Encode 3D geometry:** As is suggested in Figure 3, this procedure is basically the inverse to the generation step (equation 9):

$$\mathbf{z}_i^{(\text{node})}, \mathbf{z}_{1:i-1,i}^{(\text{bond})}, \mathbf{z}_i^{(\text{pos})} = \mathcal{F}_i^{-1}\left(\mathbf{x}_i^{(\text{node})}, \mathbf{x}_{1:i-1,i}^{(\text{bond})}, \mathbf{x}_i^{(\text{pos})}; \mathbf{e}_{1:N-1}\right) \tag{16}$$

To be specific, $\mathbf{z}_i$ is derived as follows in the training phase:

$$\mathbf{z}_i^{(\text{node})} = \left(\mathbf{x}_i^{(\text{node})} - \mu_i^{(\text{node})}\right) \odot \frac{1}{\sigma_i^{(\text{node})}} \tag{17}$$

$$\mathbf{z}_{1:i-1,i}^{(\text{bond})} = \left(\mathbf{x}_{1:i-1,i}^{(\text{bond})} - \mu_{1:i-1,i}^{(\text{bond})}\right) \odot \frac{1}{\sigma_{1:i-1,i}^{(\text{bond})}} \tag{18}$$

$$\mathbf{z}_i^{(\text{pos})} = \left(\mathbf{x}_i^{(\text{pos})} - \mu_i^{(\text{pos})}\right) \odot \frac{1}{\sigma_i^{(\text{pos})}} \tag{19}$$

where $\sigma_i$ and $\mu_i$ are derived in the same way as Equations 10, 12 and 14.

**Optimization:** At the core of the variational flow methodology is our belief that molecular data are intrinsically multi-modal, and representations of different modality should be rendered back into the same molecular entity. That is to say, whether the input data is 2D structure or 3D geometry, a well-trained model would project them to the same distribution in the chemical space. That is why in

Figure 3, we would expect the generated molecular distribution $\hat{X}_i$ to overlap with the ground-truth distribution $X_i$, even without the guidance of 2D structure $\mathcal{R}$. We formalize this intuition into a bi-level optimization task for training step $i$:

$$\text{maximize} \quad p_{X_i}(\mathbf{x}_i)\,, \tag{20}$$

$$\text{subject to} \quad D_{\text{KL}}(X_i||\hat{X}_i) < \xi \tag{21}$$

since $\mathcal{F}_i$ are invertible functions ($i = 1...N$), the LHS of Inequality 21 can be transformed into:

$$\mathcal{L}_{\text{KL}}^b = D_{\text{KL}}(X_i||\hat{X}_i) \tag{22}$$

$$= D_{\text{KL}}(\mathcal{F}_i(Z)||\mathcal{F}_i(\hat{Z})) \tag{23}$$

$$= D_{\text{KL}}(Z||\hat{Z}) \tag{24}$$

$$= D_{\text{KL}}(\mathcal{N}(\boldsymbol{\mu}_{\mathcal{R}}, \boldsymbol{\Sigma}_{\mathcal{R}})||\mathcal{N}(\mathbf{0}, \boldsymbol{I})) \tag{25}$$

where $b$ stands for a particular protein-ligand pair. This is an interesting result, which shows that across the $N$ training steps of the same protein-ligand pair, we need only control the KL divergence between $\mathcal{N}(\boldsymbol{\mu}_{\mathcal{R}}, \boldsymbol{\Sigma}_{\mathcal{R}})$ and $\mathcal{N}(\mathbf{0}, \boldsymbol{I})$. Note that $\mathcal{N}(\boldsymbol{\mu}_{\mathcal{R}}, \boldsymbol{\Sigma}_{\mathcal{R}})$ is contingent on structure $\mathcal{R}$: they are not the same across different protein-ligand pairs.

As for Equation 20, it corresponds to the flow loss term described in Equation 1. For step $i$ in pair $b$, it can be elaborated as:

$$\mathcal{L}_i^{(\text{node})} = -\log(\text{Prod}(\mathcal{N}(\mathbf{z}_i^{(\text{node})}|\boldsymbol{\mu}_{\mathcal{R}}, \boldsymbol{\Sigma}_{\mathcal{R})})) - \log(\text{Prod}(\frac{1}{\sigma_i^{(\text{node})}})) \tag{26}$$

$$\mathcal{L}_{1:i-1,i}^{(\text{bond})} = -\log(\text{Prod}(\mathcal{N}(\mathbf{z}_{1:i-1,i}^{(\text{bond})}|\boldsymbol{\mu}_{\mathcal{R}}, \boldsymbol{\Sigma}_{\mathcal{R})})) - \log(\text{Prod}(\frac{1}{\sigma_{1:i-1,i}^{(\text{bond})}})) \tag{27}$$

$$\mathcal{L}_i^{(\text{pos})} = -\log(\text{Prod}(\mathcal{N}(\mathbf{z}_i^{(\text{pos})}|\boldsymbol{\mu}_{\mathcal{R}}, \boldsymbol{\Sigma}_{\mathcal{R})})) - \log(\text{Prod}(\frac{1}{\sigma_i^{(\text{pos})}})) \tag{28}$$

$$\mathcal{L}_{\text{flow}}^{i,b} = \mathcal{L}_i^{(\text{node})} + \mathcal{L}_{1:i-1,i}^{(\text{bond})} + \mathcal{L}_i^{(\text{pos})} \tag{29}$$

Therefore, the bilevel optimization task adopts a $\beta$-regularized loss form. Note that $\mathcal{N}(\boldsymbol{\mu}_{\mathcal{R}}, \boldsymbol{\Sigma}_{\mathcal{R}})$ is involved in both loss terms:

$$\mathcal{L}_{\text{total}}^b = \frac{1}{N}(\sum_{i=1}^N \mathcal{L}_{\text{flow}}^{i,b}) + \beta\mathcal{L}_{\text{KL}}^b \tag{30}$$

We use mini-batch training, and optimize the batch loss $\sum_{b=1}^B \mathcal{L}_{\text{total}}^b$ with Adam (Kingma & Ba, 2014). We use $\beta$-annealing for adequate training of both 2D and 3D encoders, and to trade-off between binding affinity and controllability of the generated molecules. Algorithm 1 from Appendix B explains the training algorithm in more detail.

**Controllable generation:** During generation, the variational prior provides a flexible interface for controlling certain properties of the generated molecules, without the need to even re-train the model. This can be achieved by taking a collection of molecules with a certain desirable property $\rho$, denotes as $\{\mathcal{R}_a\}_{a\in I}$, where $I$ is the index set. The latent distribution for the desired property $\rho$ is defined as $\mathcal{N}(\boldsymbol{\mu}_\rho, \boldsymbol{\Sigma}_\rho)$, which can be naturally parameterized as:

$$(\boldsymbol{\mu}_\rho, \boldsymbol{\Sigma}_\rho) = \frac{1}{|I|}\sum_{a\in I}(\boldsymbol{\mu}_{\mathcal{R}_a}, \boldsymbol{\Sigma}_{\mathcal{R}_a}). \tag{31}$$

where $(\boldsymbol{\mu}_{\mathcal{R}_a}, \boldsymbol{\Sigma}_{\mathcal{R}_a})$ ($a \in I$) can be obtained from the tree encoder. Intuitively, $(\boldsymbol{\mu}_\rho, \boldsymbol{\Sigma}_\rho)$ contains inductive bias for the desired property $\rho$, and ligands that are sampled from this distribution should be more likely to possess property $\rho$.

# 5 EXPERIMENTS

## 5.1 3D MOLECULAR GENERATION CONDITIONED ON PROTEIN POCKET

**Dataset.** We use the benchmarking CrossDocked dataset (Francoeur et al., 2020), which contains 22.5 million protein-ligand pairs, to evaluate the generation performance of GraphVF. For fair comparison, we follow Pocket2Mol (Peng et al., 2022) to prepare and split the data.

| Method | HA↑ | SA↑ | QED↑ | LogP | Diversity↑ | Time↓ |
|---|---|---|---|---|---|---|
| GraphBP | 0.134 | 0.585 | 0.515 | 2.610 | 0.835 | **20** |
| Pocket2Mol | 0.272 | **0.765** | **0.563** | 1.586 | 0.688 | 2504 |
| GraphVF (w/o 2D encoder) | 0.263* | 0.542 | 0.310 | 3.567 | 0.807 | 68 |
| GraphVF (ours) | **0.311*** | 0.570 | 0.406 | **0.141** | **0.930** | 32 |

Table 2: Performance of different methods on 3D molecular generation based on protein pockets. Higher values indicate better results. Best results are in **bold**. Result with * is dealt with special valency constraints. Valency constraints are detailed in Appendix C.

**Setup.** Following GraphBP (Liu et al., 2022) and Pocket2Mol, we randomly sample 100 molecules for every protein pocket in the generation stage. The quality of generated molecules is evaluated by 6 widely adopted metrics. **(1) High Affinity (HA)**, which estimates the percentage of generated molecules that have higher *CNNAfinity* calculated by the *Gnina* program (McNutt et al., 2021); **(2) Synthetic Accessibility (SA)**, which represents the easiness of drug synthesis; **(3) Quantitative Estimation of Drug-likeness (QED)**, a measure of drug-likeness based on the concept of desirability (Bickerton et al., 2012); **(4) LopP** denotes the octanol-water partition coefficient, and good drug candidates always have a LogP between -0.4 and 5.6 (Ghose et al., 1998); **(5) Diversity** is calculated as $1 -$ average Tanimoto similarities of generated molecules for every protein pockets, following Pocket2Mol; **(6) Time** estimates the time(s) spent on generating 100 molecules for a pocket. We choose GraphBP and Pocket2Mol as our baselines, which reprent the state-of-art models for binding molecule generation. For GraphBP and GraphVF, we trained them on the dataset for 40 epochs with the same hyperparameters. For Pocket2Mol (Peng et al., 2022), we obtain the pre-trained model from their authors, and then compute the scores using *Gnina*.

**Results.** The comparison results are presented in Table 5. We can see that our GraphVF outperforms the two state-of-the-art baselines in terms of both binding affinity (HA) and Diversity. As is shown by the generation time, GraphVFis much more efficient than Pocket2Mol in 2 orders of magnitude. We have also attained comparable performance on certain properties like QED, LogP and SA, even without the explicit guidance from the variational prior. We also conduct experiment on the ablational 'w/o 2D encoder' variant in Table 5. Without the information provided by the 2D encoder, the HA value for the abalational variant drops drastically from 31.1% to 26.3%. This shows that GraphVF learns better with the 2D encoder, and significantly benefits from the variational flow architecture. We also provide visualizations of selected generated ligands in Figure 6 from Appendix E.

## 5.2 Sub-structure Analysis

**Setup.** As it is pointed out in Pocket2Mol Peng et al. (2022) that conventional metrics could not reflect the geometry of sampled molecules, we conduct additional sub-structure analysis following Pocket2Mol. For the distribution of bond angles and dihedral angles, we evaluate GraphVF by KL divergence on the same benchmarks as Pocket2Mol. We perform extra experiments on the distribution of bond length.

**Results.** The results are presented in Appendix D. In comparision to GraphBP and Pocket2Mol, GraphVF yields the best results on dihedral angles, which indicates it is more capable at modeling complex dependencies. At the same time, it achieves compatible results on bond length and bond angles.

## 5.3 Controllable Generation for Specified Chemical Sub-structures

Our pretrained framework could be used to encourage the desired sub-structures to be generated without losing diversity. We carry out case studies on generation of molecules containing the following motifs: oxhydryl, peptide bond, 6-member carbon ring, and 5-member ring containing element S. For each motif, we calculate $(\boldsymbol{\mu}_\rho, \boldsymbol{\Sigma}_\rho)$ among 500 randomly sampled reference ligand molecules that contain the motif as a sub-structure. Finally, we calculate the rate of the generated molecules that contain the desired sub-structures on the test set, which is compared with the results of directly sampling from prior distribution $\mathcal{N}(\boldsymbol{0}, \boldsymbol{I})$.

The experimental results are summarized in Table 3. With the prior distributions collected from molecules that contain certain desired sub-structures, our model is more likely to generate ligand molecules with those sub-structures.

Table 3: Controllable Generation for Specified Chemical Sub-structures.

| Rate of desired sub-structure(%) | w/ latent $\rho$ | w/o latent $\rho$ |
|---|---|---|
| oxhydryl | **51.7** | 42.8 |
| peptide bond | **6.4** | 1.5 |
| 6-member carbon ring | **14.5** | 0.3 |
| 5-member ring containing element S | **29.7** | 0.4 |

### 5.4 CONTROLLABLE MOLECULAR GENERATION FOR SPECIFIED DRUG PROPERTIES

Our framework can also be explicitly controlled to generate drug-like molecules with desired properties. To support this claim, we perform case studies under two classical pharmaceutic settings:

1) **Antibiotic Discovery** (Stokes et al., 2020) This task aims to identify molecules that inhibit the growth of *E. coli*, a bacterium canonically used for testing antibiotic activity.

2) **SARS Inhibition** (Tokars & Mesecar, 2021). This task is to identify molecules that inhibit the 3CL protease of SARS-CoV, the pathogen to a respiratory pandemic during the 2000s.

For antibiotic discovery (likewise for SARS Inhibition), the inhibition scores of all the reference ligands in the CrossDocked test set are evaluated via a pretrained ensemble model (Yang et al., 2019). We select the top 5% among them with the highest inhibition scores to calculate $(\boldsymbol{\mu}_\rho, \boldsymbol{\Sigma}_\rho)$.

Results for the two case studies are presented in Table 4, which clearly shows that the latent $\mathcal{P}$ is effective in terms of manipulating desired properties of the generated molecules.

Table 4: Controllable Generation for Antibiotic Discovery and SARS Inhibition.

| Avg. Inhibition(%) | w/ latent $\rho$ | w/o latent $\rho$ |
|---|---|---|
| Antibiotic | **3.26** | 1.03 |
| SARS | **28.3** | 11.5 |

## 6 CONCLUDING REMARKS

We proposed GraphVF, a novel variational flow-based framework for controllable binding 3D molecule generation. We empirically demonstrated that, through effectively integrating 2D structure semantics and 3D pocket geometry, GraphVF obtained superior performance to the state-of-the-art strategies for pocket-based 3D molecule generation. We also experimentally showed that, GraphVF can effectively generate binding molecules with desired ligand sub-structures and biochemical properties.

Our work here demonstrates that domain constraints can be effectively leveraged by deep generative models to improve the qualities of molecule design and fulfill the needs for controllable molecule generation. Our studies here shed light on the potential of generating binding ligands with sophisticated domain knowledge and finer-grained control over a variety of bio-chemical properties.

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

# Appendix

## A    DISCUSSIONS ON LIGAND SCAFFOLDS ENCODING

### A.1    PARSING AND POOLING

There are 3 types of canonical sub-structures, as are exemplified by the DMT molecule in Figure 1:

1) Rings, e.g. the blue, green nodes;
2) Non-ring covalent atom pairs, e.g. the red, yellow and purple nodes;
3) Pivot atoms that are connected to 3 or more items, e.g. the gray node.

The rules for identifying sub-structures is self-contained, yielding a relative sparse and stable set of vocabulary. A total of 427 canonical sub-structures are identified from the 100,000 reference ligands in the CrossDocked dataset.

Once the ligand molecule is parsed into a compilation of sub-structures, the molecular graph can be pooled into a junction tree in a straight-forward manner, where each sub-structure corresponds to a tree node, and any two intersecting sub-structures yield an edge between their corresponding nodes.

### A.2    TREE GRU ENCODER

We adopt the same junction tree encoder architecture as Jin et al. (2018). Namely, the tree message passing scheme arbitrarily selects a leaf node as the root, and pass messages from child nodes to parent nodes iteratively in a bottom-up approach. We denote the message from node $i$ to $j$ as $\mathbf{m}_{ij}$, which is updated via a GRU adapted for tree propagation:

$$\mathbf{m}_{ij} = \mathrm{GRU}(\mathbf{x}_i, \{\mathbf{m}_{ki}\}_{k \in N(i) \setminus j}) \tag{32}$$

To be more specific, the GRU architecture is formulated as follows:

$$\mathbf{s}_{ij} = \sum_{k \in N(i) \setminus j} \mathbf{m}_{ki}. \tag{33}$$

$$\mathbf{z}_{ij} = \sigma(\mathbf{W}^z \mathbf{x}_i + \mathbf{U}^z \mathbf{s}_{ij} + \mathbf{b}^z). \tag{34}$$

$$\mathbf{r}_{ki} = \sigma(\mathbf{W}^r \mathbf{x}_i + \mathbf{U}^r \mathbf{m}_{ki} + \mathbf{b}^r). \tag{35}$$

$$\widetilde{\mathbf{m}}_{ij} = \tanh(\mathbf{W}\mathbf{x}_i + \mathbf{U} \sum_{k \in N(i) \setminus j} \mathbf{r}_{ki} \odot \mathbf{m}_{ki}). \tag{36}$$

$$\mathbf{m}_{ij} = (1 - \mathbf{z}_{ij}) \odot \mathbf{s}_{ij} + \mathbf{z}_{ij} \odot \widetilde{\mathbf{m}}_{ij}. \tag{37}$$

where $\mathbf{x}_i$ is a one-hot vector, indicating the type of canonical sub-structure of node $i$. The latent representation of each node $\mathbf{h}_i$ can be derived by aggregating all the inwards messages from its child nodes:

$$\mathbf{h}_i = \mathbf{W}^o \mathbf{x}_i + \sum_{k \in N(i)} \mathbf{U}^o \mathbf{m}_{ki} \tag{38}$$

Finally, the structural encoding of the whole molecule is obtained by feeding $\mathbf{h}_{\mathrm{root}}$ through a MLP. The resulting vector $\varphi_{\mathcal{R}}$ is split into two equally-sized dense vector $(\boldsymbol{\mu}_{\mathcal{R}}, \boldsymbol{\Sigma}_{\mathcal{R}})$, which parameterize the mean and variance of an amortized diagonal Gaussian distribution $\mathcal{N}(\boldsymbol{\mu}_{\mathcal{R}}, \boldsymbol{\Sigma}_{\mathcal{R}})$:

$$(\boldsymbol{\mu}_{\mathcal{R}}, \boldsymbol{\Sigma}_{\mathcal{R}}) = \varphi_{\mathcal{R}} = \mathrm{MLP}(\mathbf{h}_{\mathrm{root}}) \tag{39}$$

It should be emphasized that all the parameters that appear in this section are **trainable**. Our earlier attempts with a pretrained version of GRU would result in a serious degradation of the quality of generated molecules. Thus, end-2-end training of these parameters is ideal for achieving better model performance.

# B  ALGORITHMS FOR TRAINING AND GENERATION

The pseudo codes of training and generation algorithms are in Algorithms 1 and 2.

---

**Algorithm 1** Training algorithm of GraphVF

---

**Input**: $\eta$ learning rate, $B$ batch size, $T$ maximum epoch number, Variational annealing hyperparameters $\beta_{\min}, \beta_{\max}$, use $\text{Prod}(\cdot)$ as the product of elements across dimensions of a tensor

**Initial**: Parameters $\theta$ of GraphVF (Echnet, Junction Tree Encoder, and Node/Edge/Position-MLP)

1: **for** $t = 1, .., T$ **do**
2:      $\beta = \beta_{\min} + (\beta_{\max} - \beta_{\min}) \sin^2\left(\pi \frac{t}{T}\right)$                $\triangleright$ $\beta$-annealing acc. to epoch number
3:      **for** $b = 1, ..., B$ **do**
4:          Sample a receptor-ligand pair from dataset, with receptor size $M$ and ligand size $N$
5:          Protein receptor $\mathcal{P} = (\tilde{V}, \tilde{E})$, where $\tilde{V} = \{(\tilde{a}_i, \tilde{r}_i)\}_{i=1}^M$, and $\tilde{E} = \{\tilde{b}_{ij}\}_{i,j=1}^M$
6:          Drug ligand $\mathcal{R} = (V, E)$, where $V = \{(a_i, r_i)\}_{i=1}^N$, and $E = \{b_{ij}\}_{i,j=1}^N$
7:          Re-order $\mathcal{R}$ with ring-first graph traversal
8:          $(\boldsymbol{\mu}_{\mathcal{R}}, \boldsymbol{\Sigma}_{\mathcal{R}}) = \text{JT-Encoder}(\mathcal{R})$, where prior $Z \sim \mathcal{N}(\boldsymbol{\mu}_{\mathcal{R}}, \boldsymbol{\Sigma}_{\mathcal{R}})$      $\triangleright$ **2D Global**
9:          **for** $i = 1, ..., N$ **do**                                  $\triangleright$ **3D Autoregressive**
10:             Construct sub-graph $\mathcal{G}_i := \tau(\mathcal{P} \cup \mathcal{R}_{1:i-1})$
11:             Construct SCS upon focal atom $f_i$
12:             $\tilde{\mathbf{e}}_{1:M}, \mathbf{e}_{1:N-1} = \text{Echnet}(\mathcal{G}_i)$                    $\triangleright$ Encode 3D conformation
13:             $\mathbf{x}_i^{(\text{node})} = a_i + \mathbf{u}, \; \mathbf{u} \sim \mathcal{U}[0,1)^d$           $\triangleright$ Atom type dequantization
14:             $\mu_i^{(\text{node})}, \sigma_i^{(\text{node})} = \text{Node-MLP}(\mathbf{e}_{f_i})$
15:             $\mathbf{z}_i^{(\text{node})} = \left(\mathbf{x}_i^{(\text{node})} - \mu_i^{(\text{node})}\right) \odot \frac{1}{\sigma_i^{(\text{node})}}$
16:             $\mathbf{x}_{1:i-1,i}^{(\text{bond})} = b_{1:i-1,i} + \mathbf{u}, \; \mathbf{u} \sim \mathcal{U}[0,1)^{(i-1) \times d}$    $\triangleright$ Bond type dequantization
17:             $\mu_{1:i-1,i}^{(\text{bond})}, \sigma_{1:i-1,i}^{(\text{bond})} = \text{Bond-MLP}(\mathbf{e}_{1:i-1}, \mathbf{x}_i^{(\text{node})})$
18:             $\mathbf{z}_{1:i-1,i}^{(\text{bond})} = \left(\mathbf{x}_{1:i-1,i}^{(\text{bond})} - \mu_{1:i-1,i}^{(\text{bond})}\right) \odot \frac{1}{\sigma_{1:i-1,i}^{(\text{bond})}}$
19:             $\mathbf{x}_i^{(\text{pos})} = \text{RBF}_{f_i}(r_i)$                          $\triangleright$ Spherize atom position to $f_i$
20:             $\mu_i^{(\text{pos})}, \sigma_i^{(\text{pos})} = \text{Position-MLP}(\mathbf{e}_{f_i}, \mathbf{x}_i^{(\text{node})}, \mathbf{x}_{1:i-1,i}^{(\text{bond})})$
21:             $\mathbf{z}_i^{(\text{pos})} = \left(\mathbf{x}_i^{(\text{pos})} - \mu_i^{(\text{pos})}\right) \odot \frac{1}{\sigma_i^{(\text{pos})}}$
22:             $\mathcal{L}_i^{(\text{node})} = -\log(\text{Prod}(\mathcal{N}(\mathbf{z}_i^{(\text{node})}|\boldsymbol{\mu}_{\mathcal{R}}, \boldsymbol{\Sigma}_{\mathcal{R}}))) - \log(\text{Prod}(\frac{1}{\sigma_i^{(\text{node})}}))$
23:             $\mathcal{L}_{1:i-1,i}^{(\text{bond})} = -\log(\text{Prod}(\mathcal{N}(\mathbf{z}_{1:i-1,i}^{(\text{bond})}|\boldsymbol{\mu}_{\mathcal{R}}, \boldsymbol{\Sigma}_{\mathcal{R}}))) - \log(\text{Prod}(\frac{1}{\sigma_{1:i-1,i}^{(\text{bond})}}))$
24:             $\mathcal{L}_i^{(\text{pos})} = -\log(\text{Prod}(\mathcal{N}(\mathbf{z}_i^{(\text{pos})}|\boldsymbol{\mu}_{\mathcal{R}}, \boldsymbol{\Sigma}_{\mathcal{R}}))) - \log(\text{Prod}(\frac{1}{\sigma_i^{(\text{pos})}}))$
25:             $\mathcal{L}_{\text{flow}}^{i,b} = \mathcal{L}_i^{(\text{node})} + \mathcal{L}_{1:i-1,i}^{(\text{bond})} + \mathcal{L}_i^{(\text{pos})}$      $\triangleright$ Step-wise loss term for normalizing flow
26:          **end for**
27:          $\mathcal{L}_{\text{KL}}^b = D_{\text{KL}}(\mathcal{N}(\boldsymbol{\mu}_{\mathcal{R}}, \boldsymbol{\Sigma}_{\mathcal{R}})||\mathcal{N}(\mathbf{0}, \boldsymbol{I}))$    $\triangleright$ Global loss term for variational distribution $Z$
28:          $\mathcal{L}_{\text{total}}^b = \frac{1}{N}\left(\sum_{i=1}^N \mathcal{L}_{\text{flow}}^{i,b}\right) + \beta \mathcal{L}_{\text{KL}}^b$
29:      **end for**
30:      $\theta \leftarrow \text{ADAM}(\sum_{b=1}^B \mathcal{L}_{\text{total}}^b, \theta, \eta)$                              $\triangleright$ Parameter update
31: **end for**

---

---

**Algorithm 2** Generation algorithm of GraphVF

---

**Input**: $T$ number of protein receptors, $B$ number of drug ligands to generate for each receptor, $N$ maximum number of atoms in the generated ligand. Optional parameters $(\boldsymbol{\mu}_\rho, \boldsymbol{\Sigma}_\rho)$ as the cue to certain desired property $\rho$, $(\mathbf{0}, \boldsymbol{I})$ by default.

**Initial**: Trained GraphVF model (Echnet, Junction Tree Encoder, and Node/Edge/Position-MLP)

---

1: **for** $t = 1, .., T$ **do**
2:      Sample a protein receptor from dataset, with receptor size $M$
3:      Protein receptor $\mathcal{P} = (\tilde{V}, \tilde{E})$, where $\tilde{V} = \{(\tilde{a}_i, \tilde{r}_i)\}_{i=1}^M$, and $\tilde{E} = \{\tilde{b}_{ij}\}_{i,j=1}^M$
4:      $\text{LigGen}_t \leftarrow [\,]$
5:      **for** $b = 1, ..., B$ **do**
6:          Drug ligand representation $\mathcal{R} := (V, E)$, initialized as empty
7:          **for** $i = 1, ..., N$ **do**
8:              Construct sub-graph $\mathcal{G}_i := \tau(\mathcal{P} \cup \mathcal{R}_{1:i-1})$
9:              $f_i = \text{FocalClassifier}(\mathcal{G}_i)$
10:             **if** none eligible for $f_i$ **then**                 ▷ Signal for generatioin complete
11:                 **break** inner loop
12:             **end if**
13:              $\tilde{\mathbf{e}}_{1:M}, \mathbf{e}_{1:N-1} = \text{Echnet}(\mathcal{G}_i)$                ▷ Encode 3D conformation
14:              Sample $\mathbf{z}_i^{(\text{node})} \sim \mathcal{N}(\boldsymbol{\mu}_\rho, \boldsymbol{\Sigma}_\rho)$
15:              $\mu_i^{(\text{node})}, \sigma_i^{(\text{node})} = \text{Node-MLP}(\mathbf{e}_{f_i})$
16:              $\mathbf{x}_i^{(\text{node})} = \sigma_i^{(\text{node})} \odot \mathbf{z}_i^{(\text{node})} + \mu_i^{(\text{node})}$           ▷ Atom type generation
17:              Sample $\mathbf{z}_{1:i-1,i}^{(\text{bond})}, \sim \mathcal{N}(\boldsymbol{\mu}_\rho, \boldsymbol{\Sigma}_\rho)$
18:              $\mu_{1:i-1,i}^{(\text{bond})}, \sigma_{1:i-1,i}^{(\text{bond})} = \text{Bond-MLP}(\mathbf{e}_{1:i-1}, \mathbf{x}_i^{(\text{node})})$
19:              $\mathbf{x}_{1:i-1,i}^{(\text{bond})} = \sigma_{1:i-1,i}^{(\text{bond})} \odot \mathbf{z}_{1:i-1,i}^{(\text{bond})} + \mu_{1:i-1,i}^{(\text{bond})}$       ▷ Bond type generation
20:              Sample $\mathbf{z}_i^{(\text{pos})} \sim \mathcal{N}(\boldsymbol{\mu}_\rho, \boldsymbol{\Sigma}_\rho)$
21:              $\mu_i^{(\text{pos})}, \sigma_i^{(\text{pos})} = \text{Position-MLP}(\mathbf{e}_{f_i}, \mathbf{x}_i^{(\text{node})}, \mathbf{x}_{1:i-1,i}^{(\text{bond})})$
22:              $\mathbf{x}_i^{(\text{pos})} = \sigma_i^{(\text{pos})} \odot \mathbf{z}_i^{(\text{pos})} + \mu_i^{(\text{pos})}$             ▷ Atom position generation
23:              Derive $(a_i, r_i)$ from $(\mathbf{x}_i^{(\text{node})}, \mathbf{x}_i^{(\text{pos})})$; $b_{1:i-1,i}$ from $\mathbf{x}_{1:i-1,i}^{(\text{bond})}$
24:              $V.\text{append}(\{(a_i, r_i)\})$; $E.\text{append}(\{b_{1:i-1,i}\})$     ▷ Autoregressive ligand generation
25:          **end for**
26:          $\text{LigGen}_t.\text{append}(\mathcal{R})$
27:      **end for**
28: **end for**
29: **return** $[\text{LigGen}_1, \text{LigGen}_2, ..., \text{LigGen}_T]$

---

## C   IMPLEMENTATION DETAILS

**Network Architecture.** We stack 6 layers of the Echnet and 20 layers of the tree GRU. We use 6 variational flow layers for generation.

**Training Details.** We train GraphVF for 40 epochs on the full training split by batch size 4. We use Adam optimizer while setting learning rate as 1e-4 and weight decay as 1e-6. For the $\beta$-annealing which is applied to the whole training process, we pick the minimum $\beta$ as 1e-4 and maximum $\beta$ as 0.015.

**Generation Details.** We sample 100 molecules for each pocket in the test set. Molecules that have less than 15 atoms are excluded and re-sampled, while molecules that have more than 50 atoms are truncated at the 50-th atom. Additionally, to help GraphVF generate ligand molecules with good geometric properties, we limit the sample space by three validity constraints during generation:

1. A bond exists between the newly generated atom and the focal atom;

2. At most one other atom could be connected to the newly generated atom with a bond.

3. The newly generated atom could only have bonds with atoms that is predicted positive by the focal classifier.

Empirically, these constraints can help. We apply all validity constraints by default for all experiments. Specially, we found that without constraint 1, GraphVF could produce better results on HA. So we removed that constraint only for that case.

# D    RESULTS ON STRUCTURAL ANALYSIS

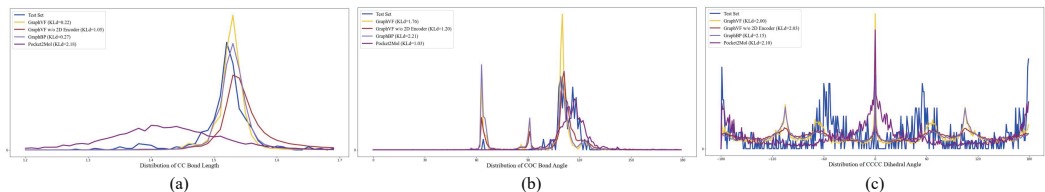

Figure 4: Visualization of bond distributions. (a) CC Bond Length; (b) COC Bond Angle; (c) CCCC Dihedral Angle.

| Sub-Structure | GraphBP | Pocket2Mol | GraphVF (w/o 2D encoder) | GraphVF |
|:---:|:---:|:---:|:---:|:---:|
| CC | 0.27 | 2.18 | 1.05 | **0.22** |
| C=O | 0.83 | 3.78 | 0.73 | **0.67** |
| CN | **0.70** | 1.78 | 1.27 | 0.77 |
| CCC | 1.31 | **1.02** | 1.07 | 1.25 |
| CCO | 0.61 | 1.38 | 0.52 | **0.51** |
| NCC | 1.00 | 1.05 | **0.62** | 0.83 |
| CCCC | 2.15 | 2.10 | 2.03 | **2.00** |
| CCCO | 2.37 | 2.27 | 2.17 | **2.17** |
| CC=CC | 2.20 | 2.85 | 2.70 | **2.04** |

Table 5: The KL divergence of the distance (upper part), bond angles (middle part) and dihedral angles (lower part) with the test set. Lowercase letters denotes atoms in aromatic rings. Best results are in **bold**.

# E  VISUALIZATION OF PROTEIN-LIGAND INTERACTIONS

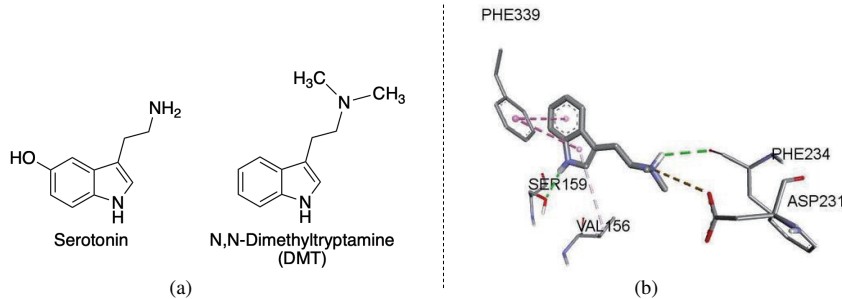

(a)    (b)

Figure 5: (a) Comparison between Serotonin and DMT structure; (b) Binding pose of DMT with 5-$HT_{2A}$, pay special attention to the interaction between $NHMe_2$ and Asp-231.

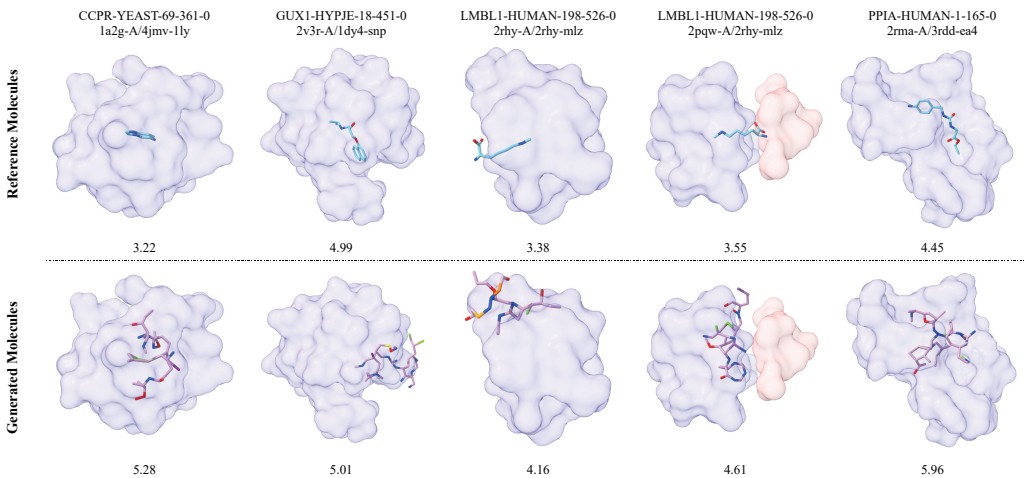

Figure 6: Examples of generated molecules with higher binding affinity (Gnina score ↑) than the reference molecules. Protein names and residue/ligand IDs are listed on top.

