# OpenReview forum: "GraphVF: Controllable Protein-Specific 3D Molecule Generation with Variational Flow"
_ICLR.cc/2023/Conference — Submitted to ICLR 2023_

### Official Review · Reviewer_uWFf · 2022-10-20

**Confidence:** 3
**Correctness:** 2
**Technical Novelty And Significance:** 2
**Empirical Novelty And Significance:** 1
**Recommendation:** 3

**Clarity, Quality, Novelty And Reproducibility:**

I found it hard to follow the description of how the model is trained. How is the loss (14) used to update parameters of the MLPs that generate atom types and bond types? (Hoogeboom et al. https://proceedings.neurips.cc/paper/2021/file/67d96d458abdef21792e6d8e590244e7-Paper.pdf seems relevant) How are reference ligands associated with training examples? Section 4 may be easier to read if the generation process is described first, and then the training procedure.

The evaluation metrics are not clearly explained (e.g. table 2). They do not assess whether generated molecules have desirable properties other than binding affinity, e.g. validity, stability, synthesisability, novelty.

I do not understand the motivation for equation (32). Would it not make more sense to sample z from a mixture of the gaussians on the right-hand side of the equation?

Code appears not to be provided.

**Strength And Weaknesses:**

The key novelty here seems to be the idea of conditioning the generation of new ligands on the coarse-grained structure of a reference ligand, or on presence of some desired substructures, by sampling the latent z from a distribution that depends on the reference ligand or on the desired substructures. This is a good idea, but the implementation is not clearly explained, and the evaluation is somewhat lacking.



**Summary Of The Paper:**

A generative model generates molecules conditioned on a protein pocket, and optionally also conditioned on a reference molecule or some desired substructures. Atom types, bonds, and atom positions are generated one at a time using an autoregressive flow model.

**Summary Of The Review:**

There is a good idea here, but the description of the implementation is hard to understand, and the evaluation is superficial.

---

> ### Author Response · Authors · 2022-11-19
> **Reply to Reviewer uWFf**
>
> We thank the reviewer for the insightful comments. Our response to the reviewer’s major concerns are listed as follows:
>
> ### 1. Organization of the paper
>
> We have followed the reviewer's advice to reorganize Section 4, which has allowed us to deliver our key methods in a much smoother way. We have also provided detailed pseudo-code of both the training algorithm and the generation algorithm, and has uploaded our source code for inspection. We hope these can help the reviewer have a firmer grasp of how our model is designed and implemented.
>
> ### 2. More evaluation metrics
> We have followed the reviewer’s advice, and have updated our experiment results on QED, LogP, SA and Diversity in Table 2. We have achieved state-of-the-art results on Gnina score and Diversity, and attained comparable performance on certain properties like QED, LogP and SA, even without the explicit guidance from the variational prior.
>
> We also perform in-depth analysis (Section 5.2) to the bond length/angle distribution of our generated molecules. The atoms and bonds we generate are more reasonably posed than the baseline models, which provides valuable insight to the superior performance of our model, and demonstrates that our result is not cherry-picked.
>
> ### 3. Technical details of variational flow
>
> #### 3.1 The loss function and parameter updates
>
> In the updated version of our paper (The 'Optimization' paragraph in Section 4.4), we have rigorously formulated how our loss term is derived. The final loss term (Equation 30):
>
> $\mathcal{L}^{b}_\mathrm{total} = \frac{1}{N}(\sum_{i=1}^{N}\mathcal{L}^{i, b}_\mathrm{flow}) + \beta \mathcal{L}_\mathrm{KL}^b$
>
> It can be exciting to find out that $\mathcal{N}(\boldsymbol\mu_\mathcal R,\boldsymbol\Sigma_\mathcal R)$ has parameterized both loss terms (flow and KL). This means that $\mathcal{N}(\boldsymbol\mu_\mathcal R,\boldsymbol\Sigma_\mathcal R)$ plays a non-trivial part in the optimization process, serving as a pivot that bridges between flow model and tree encoder.
>
> Note that reference ligand structure is the input of the tree encoder, and $(\boldsymbol\mu_\mathcal R,\boldsymbol\Sigma_\mathcal R)$ is the output of the tree encoder. The reference ligand is thus associated with the training example and other components in the training process.
>
> As for the MLPs that generate atoms types, bonds and positions, their corresponding loss terms are described in Equations 26, 27, 28.
>
> #### 3.2 Comparison with flow-based diffusion / flow-based VAE
>
> This is an interesting question in retrospect. We have inspected two related papers: flow-based diffusion (the one in the comment, Ref. 1) and flow-based VAE (Ref. 2). The workflow of Ref. 1 and Ref. 2 seem to be the opposite of our variational flow:
> 1. Variational flow uses an encoder to parameterize the prior, and directly the flow function to generate;
> 2. Flow-based Diffusion (VAE) uses the normalizing flow to parameterize the prior, and employs the diffusion-denoising (encoder-decoder) architecture to generate;
>
> Clearly, variational flow and flow based-Diffusion(VAE) are fundamentally different approaches. Variational flow is specially devised for our particular problem setting, and has been successful in combining 2D structure and 3D geometry.
>
>
> ### References
>
> [1] Hoogeboom, Emiel, et al. "Argmax flows and multinomial diffusion: Learning categorical distributions." Advances in Neural Information Processing Systems 34 (2021): 12454-12465.
> [2] Ziegler, Zachary, and Alexander Rush. “Latent normalizing flows for discrete sequences.” International Conference on Machine Learning. PMLR, 2019.

---

### Official Review · Reviewer_6f2G · 2022-10-23

**Confidence:** 5
**Correctness:** 2
**Technical Novelty And Significance:** 2
**Empirical Novelty And Significance:** 3
**Recommendation:** 3

**Clarity, Quality, Novelty And Reproducibility:**

The details about scaffold encoding needs to be better clarified. The novelty and quality of this paper is a little bit limited because the significant part of GraphVF, i.e., ligand generation process, largely follows the existing method GraphBP. Additional details such as hyperparameters are needed to increase the reproducibility.

**Strength And Weaknesses:**

Strengths:

(+) This is the work that first considers generating molecules with good drug properties in ligand generation, which is very meaningful and useful for practical applications.

(+) The effectiveness of the proposed GraphVF method is demonstrated by good performance in various experiments.

Weaknesses:

(-) (major) The paper does not clearly motivate and explain the novelty contribution of GraphVF. The design of the generation process in GraphVF (use flow models to atom-by-atom generate ligands, generate 3D positions by distances, angles, and torsion angles) largely follows the existing method GraphBP [1]. To me, the only novelty contribution is to additionally encode ligand scaffolds with junction-tree based encoders. However, it is not clear why this scaffold encoding is used and important, since the ligand generation process does not include any junction-tree generation. It is strongly recommended to add detailed discussion to clarify the difference between GraphVF and GraphBP, and the importance of using scaffold encoding.

(-) (major) It is not clear why a variational-inference-style flow model is used. As described in section 4.3 and figure 3, during training, the flow models output latent variables subjected to the Gaussian distribution $\mathcal{N}(\mu_R,\Sigma_R)$, but in loss function (equation 14-16) the latent distribution is optimized to be close to the standard Gaussian distribution, and the generation process with flow models also sample from the standard Gaussian distribution. Then what is the advantage of this method over directly modeling the outputted latent variables as the standard Gaussian distribution during training? Is it specially designed for controlled generation targeted at good drug properties? It is better to provide more detailed explanation or clarification.

(-) (major) At each generation step of GraphVF, the bonds between the new atom and previously generated atoms are generated. However, prior 3D molecule generation methods (e.g., EDM [2] and GraphBP) do not explicitly generate bonds, but infer bonds from the generated 3D conformations, and they evaluate the bond valency validity. Hence, why GraphVF does not follow this setting? Do the generated bonds match those inferred from 3D conformations? And if possible, could authors report the bond valency validity in experiments?

(-) (minor) Tables and figures in the paper do not follow some frequently used style habits in academic papers. It is better to make Figure 1, 2, 3 at the top of the page like Figure 4, make caption of Table 2 before the table, and use three-line (with \toprule and \bottomrule at the beginning and end of the table) for Table 2, 3, 4.

(-) (minor) The description about EDM in the first paragraph of section 2 is incorrect. EDM generates 3D molecular conformations from scratch, not from 2D structures.

(-) (minor) In section 3.2, use $b_i^{lig}\in\mathbb{Z}^{i-1}$ instead of $|b_i^{lig}|=i-1$ to represent an integer vector.

[1] Liu, Meng, et al. "Generating 3D Molecules for Target Protein Binding." International Conference on Machine Learning. PMLR, 2022.

[2] Hoogeboom, Emiel, et al. "Equivariant diffusion for molecule generation in 3d." International Conference on Machine Learning. PMLR, 2022.

**Summary Of The Paper:**

This paper proposes GraphVF, a novel method for generating 3D ligand molecules that can bind to specific target proteins. GraphVF adopts junction tree based scaffold encoding and SE(3)-invariant geometry encoding during training, and a variational flow framework to sequentially generate ligands atom-by-atom. GraphVF achieves good performance in ligand generation and drug property targeted generation tasks.

**Summary Of The Review:**

Though this paper presents an interesting method for controllable ligand generation and achieves good experimental performance, it has major issues concerning the motivation of using scaffold encoding, flow modeling, and bond generation. Hence, I vote for rejection and encourage authors to solve the critical issues in the revision.

---

> ### Author Response · Authors · 2022-11-19
> **Reply to Reviewer 6f2G part 1**
>
> We thank the reviewer for the extremely helpful comments. We have fixed all the minor issues mentioned in the comment. Concerning the 3 major issues, our response is listed as follows:
>
> ### 1. Core technical contributions
>
> #### 1.1 Novelty contributions
>
> Our novelty contributions are three-fold.
>
> 1. A novel variational flow-based framework to seamlessly integrate geometrical and skeletal restraints, which significantly improves the quality of the generated ligands;
> 2. Flexible interface that enables generating 3D molecules with specified chemical sub-structures or bio-chemical properties.
> 3. Comprehensive empirical studies and detailed structural analysis provides strong support to our claims.
>
> #### 1.2 Junction Tree Encoding
>
> The junction tree encoding is not intended for generating 2D scaffolds, but to help the flow model generated 3D geometry better, remedying its inefficiency in precepting 2D structural constraints. The reviewer has noticed the difference between GraphVF's training and generation procedure, since the complete 2D structure of the reference ligand is only available for during training. We have polished Section 4.4 to elucidate the connection between the training and generation procedure, and have provided detailed pseudo codes for the training algorithm and the generation algorithm in Appendix B.
>
> #### 1.3 Difference with GraphBP
>
> GraphVF and GraphBP (Ref. 1) are both flow-based models, but they are critically different in numerous technical points:
>
> 1. GraphVF explicitly generates chemical bonds in flow transformation, and adopts a bond-aware message-passing scheme (aka. Echnet). This enables flexible valency check, better perception of atomic interactions, and more reasonable delineation of molecular structures.
> 2. GraphVF uses a tree encoder to generate the prior $\mathcal{N}(\boldsymbol\mu_\mathcal R,\boldsymbol\Sigma_\mathcal R)$ on the fly (aka. amortized distribution), whereas GraphBP only supports $\mathcal{N}(\mathbf{0}, \boldsymbol{I})$, which limits model expressiveness and defies controllability.
> 3. The variational loss term is derived non-trivially from a bilevel optimization task, which is conceptually different from the flow loss.
>
> ### 2. Motivation for the variational flow architecture
>
> #### 2.1 Core Methodology
>
> At the core of the variational flow methodology is our belief that molecular data are intrinsically multi-modal, and representations of different modality should be rendered back into the same molecular entity. That is to say, whether the input data is 2D structure or 3D geometry, a well-trained model would project them to the same distribution in the chemical space. That is why in Figure 3, we would expect the generated molecular distribution $\hat{X}_i$ to overlap with the ground-truth distribution $X_i$, even without the guidance of 2D structure $\mathcal{R}$.
>
> #### 2.2 The loss terms' relations with $\mathcal{N}(\boldsymbol\mu_\mathcal R,\boldsymbol\Sigma_\mathcal R)$
>
> In the updated version of our paper (The 'Optimization' paragraph in Section 4.4), we have rigorously formulated how our loss term is derived. The final loss term (Eqaution 30):
>
> $$
> \mathcal{L}^{b}_\mathrm{total} = \frac{1}{N}(\sum_{i=1}^{N}\mathcal{L}^{i, b}_\mathrm{flow}) + \beta \mathcal{L}_\mathrm{KL}^b
> $$
>
> It can be exciting to find out that $\mathcal{N}(\boldsymbol\mu_\mathcal R,\boldsymbol\Sigma_\mathcal R)$ has parameterized both loss terms (flow and KL). This means that $\mathcal{N}(\boldsymbol\mu_\mathcal R,\boldsymbol\Sigma_\mathcal R)$ plays a non-trivial part in the optimization process, serving as a pivot that bridges between flow model and tree encoder.
>
> #### 2.3 Avoiding model degradation
>
> The reviewer has raised concerns about $\mathcal{N}(\boldsymbol\mu_\mathcal R,\boldsymbol\Sigma_\mathcal R)$ degrading into $\mathcal{N}(\mathbf{0}, \boldsymbol{I})$. In practice, we adopt a $\beta$-annealing training scheme to avoid such circumstances. During training, $\beta$ is set to be small to adequately train the tree encoder to learn structural patterns. As the training proceeds, $\beta$ becomes larger to transfer the learned structural knowledge into the flow function, to help the model generate high-quality ligands, even without the aid of the reference ligand.
>
> To empirically demonstrate our claim, we have also conducted ablation experiments by removing the 2D encoder, and fix the prior to $\mathcal{N}(\mathbf{0}, \boldsymbol{I})$. As is demonstrated in Table 2, the 'w/o 2D encoder' version performs much worse than our canonical method, which shows that our loss function and training strategy are effective in combining 2D and 3D knowledge.

---

> ### Author Response · Authors · 2022-11-19
> **Reply to Reviewer 6f2G part 2**
>
> ### 3. Benefits for generating bonds
>
> - In our Echnet, explicit incorporation of bond types during message passing is beneficial for better perception of atomic interactions and more reasonable delineation of molecular structures.
> - In the flow model, explicit generation of bond entity provides guidance for generating the position of new bonds, since bond lengths and bond angles are highly characteristic.
> - Generating bonds also means that we can perform explicit valency check. Although the generated bonds still need to be refined by 3D bond inference tools, we find this kind of valency check is still helpful in generating molecules with higher quality.
> - To empirically support our claim, we also perform in-depth analysis (Section 5.2) to the bond length/angle distribution of our generated molecules. The results show that the atoms we generate are more reasonably posed than the baseline models, and provides valuable insight to the superior performance of our model.
>
>
> ### References
>
> [1] Liu, Meng, et al. “Generating 3D Molecules for Target Protein Binding.” arXiv preprint arXiv:2204.09410 (2022)

---

### Official Review · Reviewer_Ruu5 · 2022-10-24

**Confidence:** 5
**Clarity, Quality, Novelty And Reproducibility:** It's clearly unfinished work.
**Correctness:** 1
**Technical Novelty And Significance:** 2
**Empirical Novelty And Significance:** 2
**Recommendation:** 1

**Strength And Weaknesses:**

## Major comments

1. In general, the idea is simple and easy to implement. It is a combination of JT-VAE, SchNet, Chemprop, and variational flow without much novelty.
2. There are many evaluation metrics to show, e.g. QED, SA, LogP, but only high affinity is shown in Table 2. The claim is not convincing.
3. In evaluation, the number of potential similar or duplicate molecules is not investigated. Also, the reported Pocket2Mol result is much lower than that in the original paper, which should be explained.

**Summary Of The Paper:**

This paper proposes to use generation flow to generate molecules with the control of 3D structure and 2D constraints and claims to outperform other methods.

**Summary Of The Review:**

Overall, the idae is interesting for molecule generation. However, it's clearly unfinished work and should have more comprehensive experiments to support the claim. With just one evaluation metric, the results can be questioned by the readers as cherry-picked.

---

> ### Author Response · Authors · 2022-11-19
> **Reply to Reviewer Ruu5**
>
> We thank the reviewer for the helpful comments. Our response to the reviewer's major concerns are listed as follows:
>
> ### 1. Novelty of our work
>
> Our work is indeed inspired by the aforementioned works in the comment, but it is not a simple combination/ensemble of them. In fact, its most important novelties stem precisely from the careful organization of the network architecture:
>
> 1. GraphVF is a novel variational flow-based framework that seamlessly integrate geometrical and skeletal restraints.
> 2. The special architecture of GraphVF provides a flexible interface that enables generating 3D molecules with specified chemical sub-structures or bio-chemical properties.
>
> **Echnet** is the E(3)-GNN we use in this work, designed and implemented by ourselves. The original version of Schnet (Ref. 1) does not recognize chemical bonds. We have made critical revisions to the Schnet architecture to allow for bond-aware message passing, maintaining E(3)-equivariance at the same time. (See Section 4.2: Geometry Graph Encoding)
>
> It should be emphasized that **variational flow** is not a borrowed term. As is illustrated in Figure 3, we coin this term according to the two key components in the framework: the amortized variational distribution $\mathcal{N}(\boldsymbol\mu_\mathcal R,\boldsymbol\Sigma_\mathcal R)$, and the invertible flow function $\mathcal{F}_i$. The difference between variational flow and flow-based VAE (Ref. 2) should also be highlighted:
>
> 1. Variational flow uses an encoder to parameterize the prior, and uses the normalizing flow to generate;
> 2. Flow-based VAEs uses the normalizing flow to parameterize the prior, and uses the encoder-decoder architecture to generate;
>
> Clearly, variational flow and flow based-VAE are fundamentally different approaches. Variational flow is specially devised for our particular problem setting, and has been successful in combining 2D structure and 3D geometry.
>
> ChemProp (Ref. 3) is the tool we use for validating controllable property generation in Section 5.4, and is not considered part of our model.
>
> ### 2. More evaluation metrics
>
> We have followed the reviewer's advice, and have updated our experiment results on QED, LogP, SA and Diversity in Table 2. We have achieved state-of-the-art results on Gnina score and Diversity, and attained comparable performance on certain properties like QED, LogP and SA, even without the explicit guidance from the variational prior.
>
> We also perform in-depth analysis (Section 5.2) to the bond length/angle distribution of our generated molecules. The atoms and bonds we generate are more reasonably posed than the baseline models, which provides valuable insight to the superior performance of our model, and demonstrates that our result is not cherry-picked.
>
> ### 3. Elucidation of discrepancy with Pocket2Mol on High Affinity (HA)
>
> The discrepancy is natural, because we calculate HA according to different scoring metrics: we use the Gnina (Ref. 4) metric, while Pocket2Mol (Ref. 5) uses the Vina (Ref. 6) metric. Numerous works (Ref. 7, 8) have demonstrated that Gnina score is a more reasonable metric for binding affinity, since Vina is strongly affected by the counts of rings, counts of bonds and counts of atoms.
>
>
> ### References
>
> [1] Schütt, Kristof, et al. "Schnet: A continuous-filter convolutional neural network for modeling quantum interactions." (NeurIPS 2017)
>
> [2] Ziegler, Zachary, and Alexander Rush. "Latent normalizing flows for discrete sequences." International Conference on Machine Learning. PMLR, 2019.
>
> [3] https://github.com/wengong-jin/chemprop
>
> [4] McNutt, Andrew T., et al. "GNINA 1.0: molecular docking with deep learning." Journal of cheminformatics 13.1 (2021): 1-20.
>
> [5] Peng, Xingang, et al. "Pocket2Mol: Efficient Molecular Sampling Based on 3D Protein Pockets." arXiv preprint arXiv:2205.07249 (2022).
>
> [6] Trott, Oleg, and Arthur J. Olson. "AutoDock Vina: improving the speed and accuracy of docking with a new scoring function, efficient optimization, and multithreading." Journal of computational chemistry 31.2 (2010): 455-461.
>
> [7] Liu, Meng, et al. “Generating 3D Molecules for Target Protein Binding.” arXiv preprint arXiv:2204.09410 (2022)
>
> [8] Anonymous, "Pocket-specific 3D Molecule Generation by Fragment-based Autoregressive Diffusion Models." Submitted to ICLR 2023.

---

### Official Review · Reviewer_Mkbw · 2022-10-25

**Confidence:** 3
**Correctness:** 2
**Technical Novelty And Significance:** 2
**Empirical Novelty And Significance:** 2
**Recommendation:** 3

**Clarity, Quality, Novelty And Reproducibility:**

Clarity & Quality: The paper is hard to follow. Maybe the figure description of the whole structure will be helpful. The notations and equations are obscure.

Novelty: Marginal.

Reproducibility: Difficult, the method involves too many components and most of them are less discussed.

**Strength And Weaknesses:**

Strength：

(1) Incorporating geometrical and skeletal restraints for controllable generation is novel and interesting.

(2) The experiment results demonstrate the proposed method achieves superior performance than the state-of-the-art methods.

Weaknesses:

(1) The writing of the paper is obscure. There are many notations used without explanation. The technique part of this paper is very confusing. Some equations about network structure can be put into the appendix.

(2) The technique contribution of this method is not clear. It seems the core component of the method is same as existing methods, but the difference is not highlighted.

(3) It seems that contributions 1 & 2 are the same to me. More elaborations about this are needed if I am wrong.

(4) What do graph clustering and tree encoding refer to? The construction of the mean and variance $\mu_{\mathcal{R}}, \Sigma_{\mathcal{R}}$ for target proteins is very important for the whole method but is less discussed.

(5) From Eq.14, Eq.15, and Eq.16, it seems the mean and variance $\mu_{\mathcal{R}}, \Sigma_{\mathcal{R}}$ about target proteins are fixed while optimizing the variational flow. If this is the case, the novelty of this paper would be straightforward.

(6) It seems the whole method is highly correlated to GraphBP, thus the technique difference of these two methods should be discussed.

**Summary Of The Paper:**

A controllable 3D molecule generation framework named GraohVF is proposed, in particular, the proposed GraohVF is based on variational flow and integrates geometrical and skeletal restraints for protein target generation.


**Summary Of The Review:**

Overall, this paper is hard to follow due to poor organization. Although the motivation for incorporating geometrical and skeletal restraints for controllable generation is novel and interesting, the proposed method is not technically sound. This paper should be highly polished to highlight the technique contributions and the connection of different components.

---

> ### Author Response · Authors · 2022-11-19
> **Reply to Reviewer Mkbw Part 1**
>
> We thank the reviewer for the extremely helpful comments. We have polished the whole paper to highlight our technical contributions, and remove all the obscure statements.
>
> Concerning the weaknesses mentioned by the reviewer, our response is listed as follows:
>
> ### 1. Clarity of statement
>
> We have provided the pseudo-code for both the training algorithm and the generation algorithm in Appendix B. We move detailed descriptions of the tree encoder (Section 4.1: ligand scaffold encoding) to Appendix A. Notation issues have been carefully eliminated.
>
> ### 2. Core technical contributions
>
> Our core contributions are three-fold.
>
> 1. A novel variational flow-based framework to seamlessly integrate geometrical and skeletal restraints, which significantly improves the quality of the generated ligands;
> 2. Flexible interface that enables generating 3D molecules with specified chemical sub-structures or bio-chemical properties.
> 3. Comprehensive empirical studies and detailed structural analysis provides strong support to our claims.
>
> We also need to highlight the difference between variational flow and flow-based VAE (Ref. 1):
> 1. Variational flow uses an encoder to parameterize the prior, and uses the normalizing flow to generate;
> 2. Flow-based VAEs uses the normalizing flow to parameterize the prior, and uses the encoder-decoder architecture to generate;
>
> Clearly, variational flow and flow based-VAE are fundamentally different approaches. Variational flow is specially devised for our particular problem setting, and has been successful in combining 2D structure and 3D geometry.
>
> ### 3. Distinguishing the 1st and 2nd contribution
>
> We have polished our statements about the two contributions to make them more distinguishable. Intuitively, the two contributions address two completely different problems:
> - 1st contribution: To **train** a good model that considers both 2D structure and 3D geometry. This contribution overcomes the limitations of those purely geometry-based methods like GraphBP (Ref. 2) and Pocket2Mol (Ref. 3).
> - 2nd contribution: To **generate** molecules with whatever properties we want. This contribution caters to downstream pharmaceutical tasks, where molecules of special properties are often needed to remain functional in a certain bio-chemical system.
>
> ### 4. Tree encoding
>
> We have revised Figure 2 and its related statements to avoid confusion.
>
> 1. 'Graph clustering' has been rephrased into a more precise term 'Graph pooling'. Now that the ligand molecule is parsed into a compilation of sub-structures, the molecular graph can be pooled into a junction tree in a straight-forward manner, where each sub-structure corresponds to a tree node, and any two intersecting sub-structures yield an edge between their corresponding nodes.
> 2. 'Tree encoding' refers to the process of encoding the junction tree structure via GRU message passing. The encoding is then segmented into two vectors of equal length: $\boldsymbol\mu_\mathcal R$ and $\boldsymbol\Sigma_\mathcal R$, which are used to parameterize $\mathcal N(\boldsymbol\mu_\mathcal R,\boldsymbol\Sigma_\mathcal R)$.
>
> Following the reviewer's advice, we elaborate on the whole procedure for tree encoding and construction of $(\boldsymbol\mu_\mathcal R,\boldsymbol\Sigma_\mathcal R)$ in Appendix A.
>
> ### 5. Training $(\boldsymbol\mu_\mathcal R,\boldsymbol\Sigma_\mathcal R)$ and the tree encoder
>
> Both $(\boldsymbol\mu_\mathcal R,\boldsymbol\Sigma_\mathcal R)$ and the tree encoder are trainable. Our earlier attempts with pretrained tree encoders have resulted in a serious degradation in the model performance, so fixing $(\boldsymbol\mu_\mathcal R,\boldsymbol\Sigma_\mathcal R)$ and the tree encoder during training is totally unacceptable.
>
> $(\boldsymbol\mu_\mathcal R,\boldsymbol\Sigma_\mathcal R)$ is not fixed during training: for each protein-ligand pair, it is encoded **on-the-fly**. Refer to the new equation $22\sim 30$: $(\boldsymbol\mu_\mathcal R,\boldsymbol\Sigma_\mathcal R)$ is involved in both the flow loss term and the KL loss term. It should be emphasized that optimizing $(\boldsymbol\mu_\mathcal R,\boldsymbol\Sigma_\mathcal R)$ is the pivot to mapping 2D and 3D representations to the same distribution in the chemical space, and to help the model learn better in this multi-modal setting. (See the 'Optimization' paragraph in Section 4.4 for elaboration on this point.)

---

> ### Author Response · Authors · 2022-11-19
> **Reply to Reviewer Mkbw Part 2**
>
> ### 6. Difference with GraphBP
>
> GraphVF and GraphBP (Ref. 2) are both flow-based models, but they are critically different in numerous technical points:
>
> 1. GraphVF explicitly generates chemical bonds in flow transformation, and adopts a bond-aware message-passing scheme (aka. Echnet). This enables flexible valency check, better perception of atomic interactions, and more reasonable delineation of molecular structures.
> 2. GraphVF uses a tree encoder to generate the prior $\mathcal{N}(\boldsymbol\mu_\mathcal R,\boldsymbol\Sigma_\mathcal R)$ on the fly (aka. amortized distribution), whereas GraphBP only supports $\mathcal{N}(\mathbf{0}, \boldsymbol{I})$, which limits model expressiveness and defies controllability.
> 3. The variational loss term is derivedly non-trivially from a bilevel optimization task, which is conceptually different from the flow loss.
>
> ### 7. Code Availability
>
> We have uploaded our code for inspection.
>
> ### References
>
> [1] Ziegler, Zachary, and Alexander Rush. "Latent normalizing flows for discrete sequences." International Conference on Machine Learning. PMLR, 2019.
>
> [2] Liu, Meng, et al. "Generating 3D Molecules for Target Protein Binding." arXiv preprint arXiv:2204.09410 (2022)
>
> [3] Peng, Xingang, et al. "Pocket2Mol: Efficient Molecular Sampling Based on 3D Protein Pockets." arXiv preprint arXiv:2205.07249 (2022).

---

### Author Response · Authors · 2022-11-19
**Thanks to all the reviewers**

We thank all the reviewers for their insightful comments, which have significantly helped us improve our paper. We have uploaded a new version of our paper for rebuttal revision, with 5 added pages of appendix. Important changes are highlighted as **RED**. We have also uploaded our source code for inspection, which can be downloaded in 'Supplementary Material'. We believe our revision is able to settle most concerns proposed by the reviewers. **We have also sent a dedicated reply to each reviewer, which directly answers the questions he/she has raised in the comment.**

The major improvements we have made in this revision are listed as follows:

### 1. Three-fold core technical contributions

- A novel variational flow-based framework to seamlessly integrate geometrical and skeletal restraints, which significantly improves the quality of the generated ligands;
- Flexible interface that enables generating 3D molecules with specified chemical sub-structures or bio-chemical properties;
- Comprehensive empirical studies and detailed structural analysis provides strong support to our claims.

### 2. Elaboration on ligand scaffold encoding

We have explicitly explained how the variational prior $\mathcal N(\boldsymbol\mu_\mathcal R,\boldsymbol\Sigma_\mathcal R)$ is parameterized via junction tree encoding. Details are elaborated on in Appendix A.

### 3. Reorganizing descriptions of the Variational Flow Framework

We have carefully reorganized Sections 4.3 and 4.4 for the smooth delivery of GraphVF's generation and training procedures. We have also provided pseudo-codes for both the training algorithm and the generation algorithm in Appendix B.

We coin the term 'variational flow' according to the two key components in our framework: the amortized variational distribution $\mathcal{N}(\boldsymbol\mu_\mathcal R,\boldsymbol\Sigma_\mathcal R)$, and the invertible flow function $\mathcal{F}_i$. Variational flow is well motivated to 'project data of 2D/3D modality to the same distribution in chemical space'. We have rigorously formulated the derivation of the VF loss term.

### 4. More comprehensive empirical results

We have supplemented a lot more experiments to support our claim:

- We compare the HA, SA, QED, LogP, Diversity, and Time cost of GraphVF against competitive baselines. Our model achieves SOTA on binding affinity (HA, the most important metric) and Diversity with high efficiency, and achieves comparable results with the baselines on other properties, even without explicit guidance from the variational prior.
- We provide an in-depth analysis of the bond length/angle distribution of our generated molecules and show that GraphVF can generate molecules with more reasonable structures.
- We visualize 5 generated molecules with good binding affinity to their protein receptors in Appendix E.

---

### Decision · Program_Chairs · 2023-01-20

**Decision:**

Reject

**Justification For Why Not Higher Score:**

All reviewers are concerned with many aspects of this paper, including methods and innovations, writing, and experiments. Thus a reject is recommended.

**Justification For Why Not Lower Score:**

NA

**Metareview: Summary, Strengths And Weaknesses:**

This paper studies 3D molecule generation conditional binding proteins. All reviewers are concerned with many aspects of this paper, including methods and innovations, writing, and experiments. Thus a reject is recommended.